# Contribution of genetic factors to high rates of neonatal hyperbilirubinaemia on the Thailand-Myanmar border

Germana Bancone [1,2]*, Gornpan Gornsawun[1], Pimnara Peerawaranun[3], Penporn Penpitchaporn[1], Moo Kho Paw[1], Day Day Poe[1], December Win[1], Naw Cicelia[1], Mavuto Mukaka[2,3], Laypaw Archasuksan[1], Laurence Thielemans[4], Francois Nosten[1,2], Nicholas J. White[2,3], Rose McGready[1,2], Verena I. Carrara[1,2,5]

1 Shoklo Malaria Research Unit, Mahidol-Oxford Tropical Medicine Research Unit, Faculty of Tropical Medicine, Mahidol University, Mae Sot, Thailand, 2 Centre for Tropical Medicine and Global Health, Nuffield Department of Medicine, University of Oxford, Oxford, United Kingdom, 3 Mahidol-Oxford Tropical Medicine Research Unit (MORU), Faculty of Tropical Medicine, Mahidol University, Bangkok, Thailand, 4 Neonatology-Pediatrics Department, Université Libre de Bruxelles, Bruxelles, Belgium, 5 Institute of Global Health, Faculty of Medicine, University of Geneva, Geneva, Switzerland

* germana@tropmedres.ac

**Data Availability Statement:** There are ethical restrictions on sharing data publicly and we do not have consent from participants for their data to be

## Abstract

Very high unconjugated bilirubin plasma concentrations in neonates (neonatal hyperbilirubinaemia; NH) may cause neurologic damage (kernicterus). Both increased red blood cell turn-over and immaturity of hepatic glucuronidation contribute to neonatal hyperbilirubinaemia. The incidence of NH requiring phototherapy during the first week of life on the Thailand-Myanmar border is high (approximately 25%). On the Thailand-Myanmar border we investigated the contribution of genetic risk factors to high bilirubin levels in the first month of life in 1596 neonates enrolled in a prospective observational birth cohort study. Lower gestational age (<38 weeks), mutations in the genes encoding glucose-6-phosphate dehydrogenase (G6PD) and uridine 5′-diphospho-glucuronosyltransferase (UGT) 1A1 were identified as the main independent risk factors for NH in the first week, and for prolonged jaundice in the first month of life. Population attributable risks (PAR%) were 61.7% for lower gestational age, 22.9% for hemi or homozygous and 9.9% for heterozygous G6PD deficiency respectively, and 6.3% for UGT1A1*6 homozygosity. In neonates with an estimated gestational age ≥ 38 weeks, G6PD mutations contributed PARs of 38.1% and 23.6% for "early" (≤ 48 hours) and "late" (49–168 hours) NH respectively. For late NH, the PAR for UGT1A1*6 homozygosity was 7.7%. Maternal excess weight was also a significant risk factor for "early" NH while maternal mutations on the beta-globin gene, prolonged rupture of membranes, large haematomas and neonatal sepsis were risk factors for "late" NH. For prolonged jaundice during the first month of life, G6PD mutations and UGT1A1*6 mutation, together with lower gestational age at birth and presence of haematoma were significant risk factors. In this population, genetic factors contribute considerably to the high risk of NH. Diagnostic tools to identify G6PD deficiency at birth would facilitate early recognition of high risk cases.

shared openly without any restrictions, and both Mahidol and Oxford Ethics Committees have agreed upon those terms. The population we work with has been displaced by conflict or work in Thailand without any legal status; their situation has become worse since the recent coup and any data pertaining to people on the border who have no legal status is therefore extremely sensitive. The data is available on request from scientists with a genuine interest in neonatal jaundice, which can be easily verified by the Mahidol Oxford Tropical Medicine Data Access Committee established in 2016, for data sharing purposes. Data are available from MORU Tropical Health Network upon request from this link: https://www.tropmedres.ac/units/moru-bangkok/bioethics-engagement/data-sharing.

**Funding:** This research was funded in part by the Wellcome Trust [grant 106698]. LT was supported by a PhD grant from "The Belgian Kids Fund for Pediatric Research". The funders had no role in study design, data collection and analysis, decision to publish, or preparation of the manuscript.

**Competing interests:** The authors have declared that no competing interests exist.

## Introduction

Neonatal hyperbilirubinaemia (NH) is common. Although it is usually benign and resolves in the first week of life without treatment, sustained very high plasma concentrations of unconjugated bilirubin are neurotoxic and cause kernicterus [1]. Morbidity and mortality from severe NH occurs predominantly in resource-limited settings as a result of delays in diagnosis and treatment [2]. Permanent sequelae to the nervous system of surviving neonates cause substantial morbidity to the affected individual and difficulties for their family. Most low-income countries have little or no infrastructure for social and medical support of affected children [3]. Identifying newborns at risk of severe NH is important therefore to permit preventive steps.

Genetic factors predisposing to haemolysis or reduced bilirubin conjugation predispose to NH [4]. X-linked glucose-6-phosphate dehydrogenase (G6PD) deficiency is the most common human enzymopathy, with an allelic frequencies averaging 8–10% in tropical areas, but in some populations reaching over 30% [5]. G6PD deficiency is expressed completely in the red cells of hemizygote males and homozygote females but, because of Lyonisation, heterozygotes have a range of phenotypic expression between deficient and normal. The increased risk of NH in G6PD deficient neonates probably results from the shortened erythrocyte lifespan, sometimes exacerbated by exposure to oxidising agents. Over 200 mutations causing reduced enzymatic activity have been described [6], affecting over 400 million people worldwide. Mahidol (487G>A), Viangchan (871G>A), Union (1360C>T), Canton (1376G>T) and Kaiping (1388G>A) are the most common variants found in the Greater Mekong Subregion [7]. These variants are historically classified as moderate to severe and can be associated with severe acute haemolysis upon exposure to oxidants.

The uridine diphosphate glucuronosyltransferase (UGT) enzymes are a superfamily of conjugating enzymes. UGT1A1 is the sole enzyme responsible for the metabolism of bilirubin. Reduced activity is associated with neonatal unconjugated hyperbilirubinemia, Gilbert's syndrome, and both type I and type II Crigler-Najjar syndromes. Several mutations cause reduced activity in the UGT1A1 protein. In the promoter region, the UGT1A1*28 and UGT1A1*37 alleles have 7 and 8 repetitions of the (TA) box respectively which impair efficient transcription, resulting in >70% reduction in gene transcription [8–10]. In the coding region, the UGT1A1*6 allele (Arg71Gly; 211G>A; rs4148323) results in a critical reduction in enzymatic activity in both homozygotes (32% of normal) and heterozygotes (60% of normal [11]). The prevalence of UGT1A1*28 is around 30% in Caucasians, between 40% and 56% in African Americans, and less than 15% in Asian populations [12]. UGT1A1*6 has been found mostly in Asian population where its allele frequency ranges from 13% to 23% [13].

Haemoglobinopathies are also potential risk factors for NH, notably in neonates born from mothers carrying sickle cell [14], or thalassaemia genetic polymorphisms.

On the Thailand-Myanmar border NH requiring phototherapy is common. G6PD deficiency was identified as a major contributory factor a decade ago [15]. A further prospective birth cohort from the same site (ClinicalTrials.gov Identifier: NCT02361788) described the epidemiology of NH and confirmed increased the risk in G6PD deficient neonates [16]. The current analysis of the same cohort assessed the relative contributions of genetic traits including G6PD and UGT1A1 mutations and maternal abnormal haemoglobins to NH.

## Materials and methods

### Study

This prospective observational birth cohort study was conducted on the Thailand-Myanmar border in three SMRU clinics between January 2015 and May 2016 (ClinicalTrials.gov

Identifier: NCT02361788). SMRU clinics serve a refugee and migrant population mainly comprising subjects of Sgaw Karen, Burman and Poe Karen ethnicities. Antenatal care (ANC) is provided free of charge. Estimation of gestation by ultrasound is routine, as are laboratory analyses including regular assessments of haematocrit concentration and malaria smear. In addition, a maternal complete blood count at the first ANC visit was performed together with a G6PD qualitative test and haemoglobin typing [17] during the study period. All live born neonates with estimated gestational age (EGA) ≥ 28 weeks were included if they were seen within 48 hours of life, or if they presented with jaundice within their first week of life. Clinical examination and laboratory tests were scheduled at defined time-points (see later) during the first week of life and weekly until one month of age [18]. Mothers were encouraged to bring their jaundiced or unwell neonates to the clinics any time in-between appointments for examination and treatment.

Total serum bilirubin (TSB) levels were used to define NH using EGA and neonatal age-adjusted treatment thresholds for phototherapy which followed NICE guidelines [19], e.g. newborns with EGA ≥38 weeks and a TSB of 260umol/L at 48 hours of life, would be diagnosed with NH and treated with phototherapy. Two types of bulbs for phototherapy were available: Phillips TL20/52 blue light bulbs of 400 to 500 nm wavelength and LED bulbs (peak wavelength 455 nm). The blue-light bulbs were either inserted into a wooden or a metallic framed cot; LED bulbs units were mobile and set directly above the baby cot. Phototherapy units delivered recommended minimal irradiance levels of 8–10 $\mu W/cm^2/nm$; the distance between the light and the cot was adjustable in order to obtain, if necessary, intensive phototherapy (≥30 $\mu W/cm^2/nm$, [20]). The first phototherapy units were set up in the clinics in 2009 [15] and by the time of this study they were well accepted by the mothers who could sleep near the cot, breastfeed, and care for their newborn.

For the analysis in neonates with EGA≥38 weeks, NH diagnosed within the first 48 hours of life was categorised as 'early NH', while NH occurring between 48-168h of life was defined as 'late NH'. The 48h cut-off was based on the median duration stay in the postnatal ward following an uncomplicated delivery in this setting. NH was defined as severe if at least one TSB was on or above the NICE-defined exchange transfusion threshold. Care for each newborn with severe NH was based on clinical assessment and TSB trajectories after diagnosis; it also included discussion with the local Thai hospital (located approximately 1-hour drive from SMRU clinics) where exchange transfusion was available. By protocol, follow up measurements included TSB, haematocrit, and daily weight for 3 days and at day 7 on all newborns. Follow-up was deemed 'complete' if a minimum of three TSB measurements were available: I) one before or at 30h hours of life, II) a second ≤36h after the first, and III) a third between 5 and 7 days of life.

Neonates were then assessed weekly until one month of age. Each visit included a clinical examination, weighing and visual assessment of jaundice. Clinically apparent jaundice assessed at any follow-up visit in neonates older than 14 days was defined as prolonged jaundice. Onsite TSB levels were checked at each visit while direct and indirect bilirubin concentrations were measured at weeks 3 or 4.

## Laboratory evaluations

G6PD status was assessed initially on cord blood by the qualitative fluorescent spot test (FST, R&D Diagnostic, Greece). ABO and Rhesus blood grouping was performed using the agglutination method with anti-A, anti-B and anti-D sera (Plasmatec, UK). TSB and haematocrit measurements were performed in centrifuged capillary heel prick samples (3 min centrifugation at 10,000 rotations per minute). Haematocrit was estimated using a Hawksley micro-

haematocrit reader. The sample were then used to assess total serum bilirubin photometrically using the Bilimeter2 or Bilimeter3 micro-bilirubinometers (Pfaff Medical GmbH, Germany). During the follow-up visits after three weeks of life, when clinically indicated, serum direct and indirect bilirubin measurements were assessed biochemically at an external accredited laboratory.

At the central haematology laboratory, newborns' DNA was extracted using column kits (Favorgen Biotech Corp., Taiwan) from 200 μL of cord blood. G6PD genotyping for Mahidol (487G>A), the most common local variant, was performed on all samples; genotyping for the other 4 local G6PD variants, Union (1360C>T), Canton (1376G>T), Kaiping (1388G>A) and Chinese-4 (392G>T) was performed only on FST-deficient samples; established protocols were used [21, 22]. Since over 90% of G6PD mutations in this population are Mahidol variant, for the statistical analyses all detected mutations were pooled; for the analyses of risk, hemizygote and homozygote genotypes were pooled. Genotyping for UGT1A1*6 (211G>A) and for TA repeats in the UGT1A1 gene promoter (UGT1A1*28, UGT1A1*26, UGT1A1*37) was adapted from published protocols and summarized in S1 Table. For the statistical analyses of risk, heterozygote and homozygote UGT1A1*28 genotypes were pooled together.

Haemoglobin typing of the mother was carried out by Capillary Electrophoresis using a Capillarys II (Sebia, France) on blood collected at the first ANC visit. Capillary Electrophoresis allows for diagnosis of Hb structural variants such as HbE, HbC, HbS (by appearance of retention peaks at specific elution times), presumed beta-thalassaemia carriage (by increased percentage of $HbA_2$), and presumptive diagnosis of alpha-thalassaemia trait (by decreased percentage of $HbA_2$). For the statistical analysis, women were classified based on the likely expected haematologic picture associated with the globin variant; normal women were grouped with carriers of presumptive alpha-thalassemia trait or HbE trait in the "Non-clinically significant haemoglobinopathies" group. Homozygous HbEE and women with beta-thalassaemia trait, and HbE/beta-thalassaemia were pooled in the "Haemoglobinopathies" group.

## Statistical analysis

The prospective observational cohort study that was used for analyses included 1,710 neonates. In order to evaluate the contributions of G6PD and UGT1A1 genotypes, and maternal abnormal haemoglobin types to the risk of NH in the first week and in prolonged jaundice, the analysis included variables related to the mother, the obstetric history, the neonate and the perinatal period previously identified in the same cohort [16]. These were maternal age, literacy, smoking, gravida, overweight (body mass index $\geq 27.5$ mg/kg$^2$ within 2 weeks of delivery [23]), pre-eclampsia or eclampsia for the mother; prolonged rupture of membranes, oxytocin infusion, delayed cord clamping for the obstetric history, gestational age, resuscitation, presence of haematoma, ethnicity, sex, size for gestational age, siblings with history of jaundice, use of naphthalene for storing clothes, G6PD deficiency by FST, potential ABO incompatibility (i.e. mother with blood group O and neonate with either A, B or AB), positive Coombs test for the neonate; and severe infection, weight loss >7%, haematocrit level and polycythaemia for the clinical events within the first 24 hours of life.

For the neonates' genotypes, allelic frequencies (p) were calculated as the total number of mutated alleles observed as a proportion of the total analysed; for G6PD mutations, males provide 1 allele per person and females provide 2. 95% CI were calculated as 1.96 multiplied by the square root of [p (1-p)]/N where N was the total number of alleles analysed, where 1.96 is the standard normal z-value corresponding to the 95% CI. Allelic frequencies were compared between ethnic groups using the Chi squared test. Neonates' ethnicity (Sgaw Karen, Poe Karen, Burman, "Burmese Muslim" and others) was based on self-reported ethnicity of both

parents and grandparents. People of Islamic faith self-identified as "Burmese Muslim" [17]. Ethnicity was reported as "mixed" when parents' ethnicity differed.

A mixed effects Cox proportional hazard model that accounted for clustering by site was used to analyse risk factors for NH in the first week of life. Accounting for clustering was important because members of the same cluster (site) tend to have more correlated outcomes compared to members of a different cluster (site). Failure to account for these correlations tends to bias p-values downwards thereby increasing type I error. The hazard ratios (HRs) and the corresponding 95% CIs from this model have been presented. Harrell's C statistic was used for Cox regression model discrimination. Because neonates born earlier have an increased risk of NH and NICE guidelines for starting treatment propose lower thresholds with each gestational week below 38 weeks, analysis of "early" and "late" NH was carried out only on newborns with EGA≥38 weeks who would normally be discharged from clinics around two days of life. In order to assess the impact of the risk factors on neonatal hyperbilirubinaemia the Population Attributable risk (PAR) percentages have been used. The PAR percentages were calculated for all significant risk factors of the multivariable analysis as: PAR% = [prevalence of exposed x (AHR-1)] / (1 + [prevalence of exposed x (AHR-1)]) x 100. The 95%CIs of PAR% were calculated using the same formula whereby AHR is replaced by the lower and upper 95% CI limits of AHR.

A mixed effects logistic model that accounted for clustering by site was used to analyse the risk of prolonged jaundice. The odds ratios (ORs) and the corresponding 95% CIs from this model are presented. The PARs for the odds ratios were also calculated using the same formula as that for AHR using AOR instead of AHR.

A mixed effects negative binomial model that took into account clustering by site was used to analyse the duration of prolonged jaundice. The incidence rate ratios (IRRs) and the corresponding 95% CIs from this model are reported. A mixed effects linear regression clustering by site was used to analyse interactions between G6PD and UGT1A1 genotypes on TSB levels. The slope and the corresponding 95% CIs from this model are reported. Comparison of total and indirect levels of bilirubin at week 3 among different genotypes was analysed by ANOVA. All tests of significance were performed at 5% level. Data were analysed using SPSS version 27 and Stata MP version 16.

### Ethics approval

The study was approved by Oxford Tropical Research Ethics Committee, UK (OxTREC 41–144), the Mahidol University Faculty of Tropical Medicine Ethical Committee, Thailand (TMEC 14–012) and the Tak Province Border Community Ethics Advisory Board (TCAB-08-13). Written informed consent was obtained from literate parents or guardians of the neonates; a thumbprint was obtained in the presence of a literate witness for illiterate parents.

### Results

The full cohort included 1,710 neonates (890 males and 820 females); a small percentage (1.2%, N = 20) were twins and were excluded from the genetic analysis because related. Twins were also excluded from the risk analysis because they are often born smaller and earlier independently from their genetic background or other clinical factors. Of the remaining 1,690 neonates, there were 120 that could not be genotyped so a total of 1,570 neonates were analysed for distribution of genetic variants among ethnic groups. The study flow is represented in Fig 1. Among the 1,690, 420 with incomplete TSB follow-up were excluded from the risk factors analysis of NH. The remaining 1,270 neonates were assessed for NH, 96.5% (1,225/1,270) of whom had an available genotype for G6PD and/or UGT1A1. Sub-analysis of early NH (within

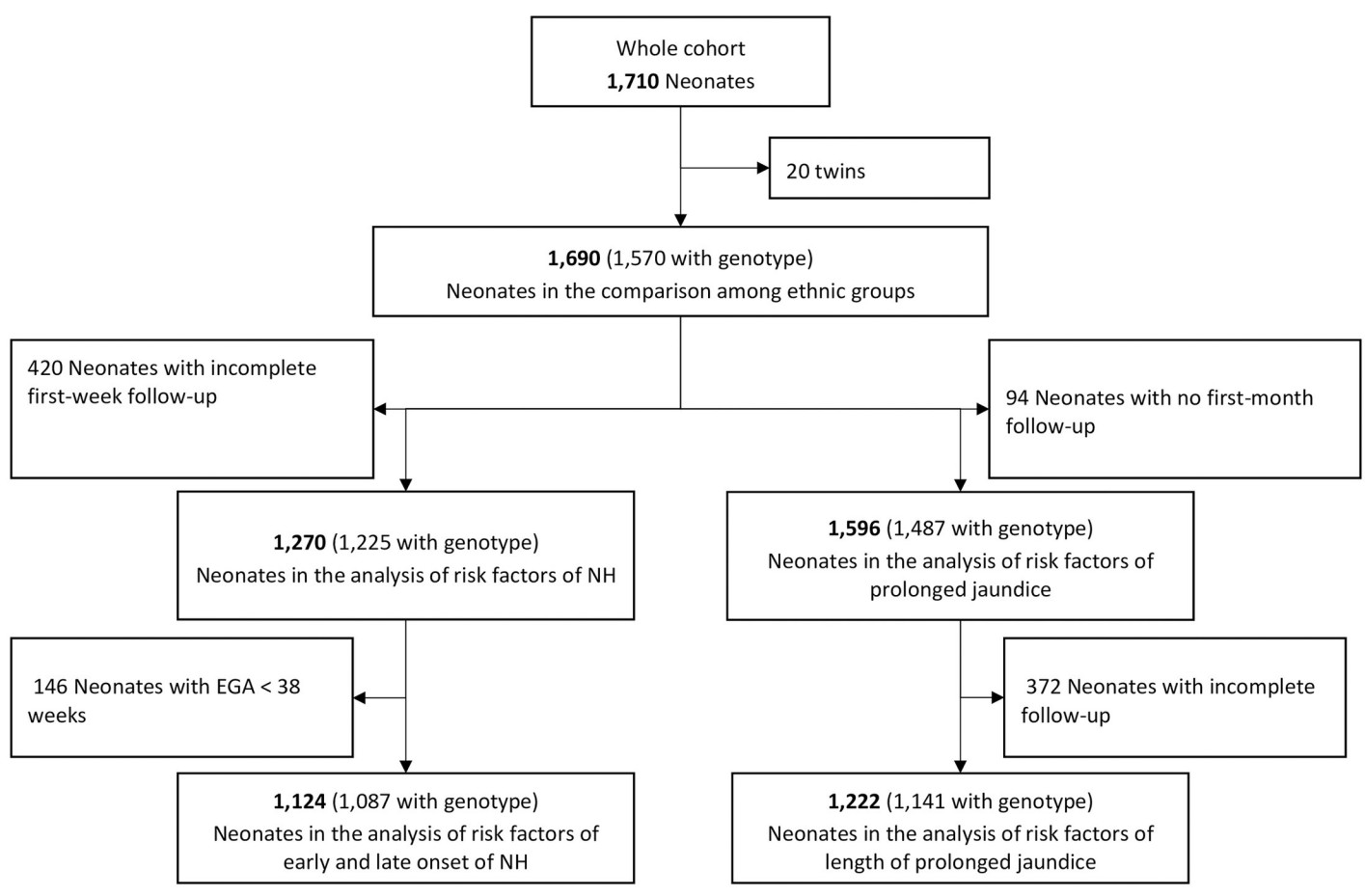

**Fig 1. Study flow.**

48h of birth) and late NH (after 48h) in the first week of life was performed in a total of 1,124 neonates born with EGA≥ 38 weeks (1,087 with a genotype; 562 males and 525 females). Prolonged jaundiced was analysed in 1,596 newborns with at least one follow-up visit of the full cohort. Analysis of the duration of prolonged jaundice was performed on 1222 neonates with full follow-up until 35 days of life.

In total 24.3% (309/1,270) neonates developed NH in the first week of life. Among the neonates with EGA≥38 weeks, 5.4% (61/1,124) developed NH early (within 48 hours) and 11.4% (128/1,124) developed NH late (49–168 hours). Among neonates with at least one follow-up until 35 days of life, 38.5% (615/1,596) had prolonged jaundice.

## G6PD and UGT1A1 genotypes

Among the 802 males genotyped for G6PD mutations, 97 (12.1%) were hemizygotes (91 Mahidol, 4 Canton and 2 Union mutations) and 7 other males were deficient by G6PD testing but none of the tested mutations were found. Among the 767 females genotyped, 10 (1.3%) were homozygotes and 156 (20.3%) were heterozygotes for the Mahidol mutation, and 1 had a deficient phenotype but no mutation was identified. The overall allelic frequency of all characterised G6PD deficient mutations was 11.6%.

Among the 1,570 neonates genotyped for the UGT1A1*6 allele, 47 were homozygotes and 440 were heterozygotes. The overall allelic frequency was 17.0%. Among the neonates genotyped for the UGT1A1 promoter (1,246), the allelic frequency of the TA7 repeat (UGT1A1*28) was 12.3%; there were 251 heterozygotes and 28 homozygotes. No TA8 repeat (UGT1A1*37) was observed in the population.

Results of genotyping by ethnic group are shown in Table 1 and Fig 2. There was a distinct association of genotypes with ethnic groups. G6PD deficient mutations were more common among newborns of Karen ethnicity (13.5%) as compared to Burman (9.2%, P = 0.011). The allelic frequency of the UGT1A1*6 mutation was significantly higher in Sgaw Karen (21.0%) as compared to Burmans (12.4%, P<0.001) and was twice as high as in "Burmese Muslims" (8.4%, P<0.001). Poe Karen (16.2%) also had a significantly higher allelic frequency of the of the UGT1A1*6 mutation compared to "Burmese Muslims" (P<0.020). Allelic frequency of UGT1A1*28 had the opposite distribution, with a significantly higher frequency in "Burmese Muslims" (28.1%) and Burmans (16.2%) as compared to Sgaw Karen (8.8%, P<0.001) and Poe Karen (9.2%, P<0.001 for "Burmese Muslims" and P = 0.025 for Burmans).

## Analysis of risk factors for NH in the first week of life

Independent of maternal, obstetric and neonatal risk factors, G6PD deficiency hemizygotes or homozygotes had an adjusted Hazard Ratio (AHR) of 4.78 (95%CI:3.35–6.84; P<0.001) for developing NH in the first week of life compared to G6PD wild type genotypes (Table 2, and S2 Table). This confirmed the results obtained previously with the G6PD FST phenotypic screening test in the same cohort [16]. In addition, females heterozygous for G6PD deficient alleles had an AHR of 2.09 (95%CI: 1.41–3.12; P<0.001) for developing NH in the first week of

**Table 1. Estimated allelic frequencies [95% confidence intervals] of genetic traits analysed in the cohort of newborns.**

| Newborn | G6PD all | p-value | UGT1A1*6 | p-value | UGT1A1*28 | p-value |
|---|---|---|---|---|---|---|
| ethnicity | mutations | | | | | |
| | MAF%[95%CI] | | MAF%[95%CI] | | MAF%[95%CI] | |
| | (N) | | (N) | | (N) | |
| Sgaw Karen | 13.5[11.3–15.8] | Reference | 21.0[18.7–23.3] | Reference | 8.8[7.0–10.5] (508) | <0.001 |
| | (594) | | (595) | | | |
| Burman | 9.2[6.6–11.9] | 0.011 | 12.4[9.8–15.0] | <0.001 | 16.2[12.7–19.7] | 0.0016 |
| | (302) | | (303) | | (216) | $P_{vs\ SK}$<0.001 |
| | | | | | | $P_{vs\ PK}$ = 0.025 |
| Poe Karen | 10.0[5.5–14.5] | 0.207 | 16.2[11.4–21.0] | 0.100 | 9.2[4.9–13.5] | <0.001 |
| | (114) | | (114) | | (87) | |
| "Burmese Muslim" | 9.7[4.5–14.9] | 0.231 | 8.4[4.2–12.7] | <0.001 | 28.1[20.8–35.4] | Reference |
| | (83) | | (83) | | (73) | |
| Others | 8.7[0.0–20.2] | NA | 17.9[3.7–32.0] (14) | NA | 10.0[0.0–23.1] (10) | NA |
| | (14) | | | | | |
| Mix | 11.5[9.0–14.0] | NA | 16.9[14.3–19.4] | NA | 12.6[10.0–15.2] | NA |
| | (413) | | (412) | | (314) | |
| Total& | 11.6[10.3–12.9] | | 17.0[15.7–18.3] | | 12.3[11.0–13.6] | |
| | (N = 1569) | | (N = 1570) | | (N = 1246) | |

MAF: minor allele frequency; NA: non analysed.

Allelic frequencies of neonates of other ethnic groups (pooled in "Others") and neonates of mixed ethnic origin were not compared to the others.

&Includes also neonates of unknown ethnicity (N = 49).

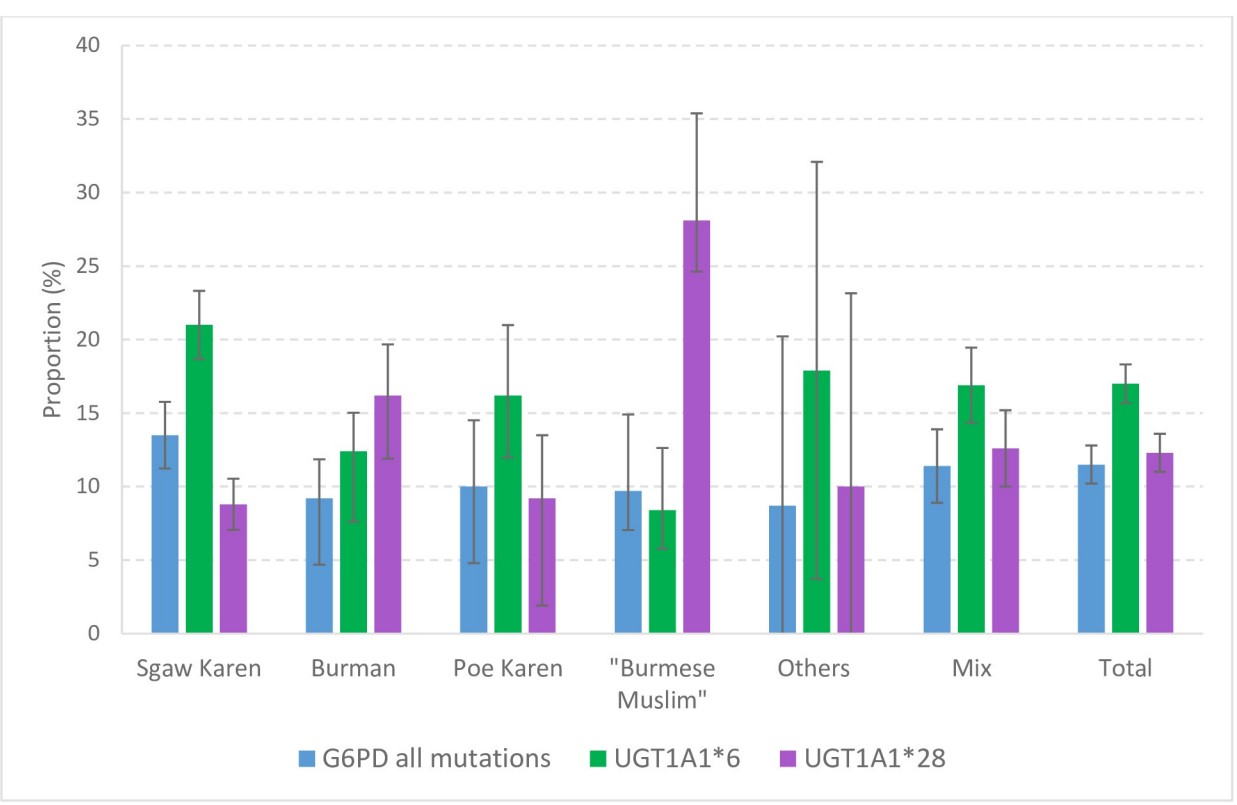

**Fig 2. Allelic frequencies of analysed genetic traits by ethnicity.** Bars indicate 95%CI.

life. Nearly all G6PD heterozygous neonates (121/123) had a "normal" phenotype assessed by the FST, and would therefore not be considered at risk of NH if a qualitative screening test only had been used. The overall PARs for G6PD hemi/homozygotes and heterozygotes compared to G6PD wild type genotype were 22.9% and 9.9% respectively.

UGT1A1*6 homozygotes had an AHR of 3.22 (95%CI:1.94–5.37; P<0.001) contributing a PAR of 6.3% for NH, but for heterozygotes the risk was not significantly increased; AHR 1.24 (95%CI:0.92–1.66; P = 0.151). Those with the UGT1A1*28 allele had a non-significant reduced AHR of 0.77 (95%CI:0.53–1.11;) when pooling heterozygous and homozygous genotypes compared to wild type genotype. Harrell's C statistic for model discrimination indicated that 81.8% of NH in the first week of life was explained by the analysed factors in the multivariable analysis.

## Severe NH

A total of 20 severe cases of neonatal jaundice in this cohort of 1,710 neonates have been described previously [16]. The current analysis of genotypes showed that a boy born at home who was seen at day 2, had fever, clinical signs of sepsis and died shortly afterwards, was hemizygous for Canton mutation (he tested G6PD deficient by FST). In two neonates who received exchange transfusion, one was a UGT1A1*6 heterozygous and G6PD*Mahidol hemizygous boy and the other was a UGT1A1*6 heterozygous and G6PD*Mahidol heterozygous girl.

Among the 1,270 neonates with full follow-up in the first week of life analysed here, 15 reached TSB levels above the exchange transfusion threshold; 5 in the group with EGA<38 weeks and 10 in the group with EGA≥38 weeks. Among the neonates with EGA≥38 weeks

**Table 2. Neonatal hyperbilirubinaemia (NH) in the first week of life: Uni- and multivariable analyses of genetic factors and other significant risk factors for developing NH using a mixed effects Cox proportional hazard model clustering by site.**

| Characteristics | Univariable analysis | | Multivariable analysis[a] | | PAR |
|---|---|---|---|---|---|
| | HR (95% CI) | p-value | HR (95% CI) | p-value | % (95% CI) |
| **Newborn genotyping** | | | | | |
| G6PD (any mutation) | | | | | |
| WT | Reference | | Reference | | |
| Heterozygote | 1.76 (1.25, 2.47) | 0.001 | 2.09 (1.41, 3.12) | <0.001 | 9.9 (4.0–17.6)[b] |
| Hemi + Homozygote | 3.58 (2.63, 4.86) | <0.001 | 4.78 (3.35, 6.84) | <0.001 | 22.9 (15.6–31.4)[b] |
| UGT1A1*6 | | | | | |
| WT | Reference | | Reference | | |
| Heterozygote | 1.18 (0.91, 1.51) | 0.206 | 1.24 (0.92, 1.66) | 0.151 | |
| Homozygote | 2.50 (1.55, 4.01) | <0.001 | 3.22 (1.94, 5.37) | <0.001 | 6.3 (2.8–11.7) |
| UGT1A1*28 | | | | | |
| WT (TA6/6) | Reference | | Reference | | |
| Hetero and homozygote (TA6/7+ TA7/7) | 0.67 (0.48, 0.95) | 0.022 | 0.77 (0.53, 1.11) | 0.160 | |
| **Maternal Characteristics** | | | | | |
| Primigravida (Primipara) | 1.81 (1.45, 2.27) | <0.001 | 1.71 (1.30, 2.25) | <0.001 | 19.6 (9.3–30.0) |
| **Obstetric characteristics** | | | | | |
| Rupture of membranes ≥ 18h | 1.76 (1.21, 2.57) | 0.003 | 2.30 (1.50, 3.53) | <0.001 | 7.9 (3.2–14.3) |
| **Neonatal Characteristics** | | | | | |
| Gestational age (<38 weeks) | 12.6 (10.0, 15.8) | <0.001 | 15.0 (11.3, 20.0) | <0.001 | 61.7 (54.2–68.6) |
| Presence of haematoma | 2.24 (1.45, 3.46) | <0.001 | 1.98 (1.18, 3.32) | 0.010 | 3.8 (0.7–8.5) |
| Sgaw Karen ethnicity | 1.36 (1.04, 1.77) | 0.025 | 1.29 (0.95, 1.74) | 0.104 | |
| Potential ABO incompatibility | 1.32 (0.98, 1.76) | 0.068 | 1.30 (0.93, 1.82) | 0.131 | |

WT: wild type; EGA: Estimated Gestational Age; PAR: population attributable Risk (sum of PAR can be >100% [24]); HR: Hazard ratio; CI: confidence interval.

[a]Adjusted for Primigravida, Pre-ecl ampsia or eclampsia, Rupture of membranes ≥ 18h, Delayed cord clamping, Gestational age <38 weeks, Presence of haematoma, Sgaw Karen ethnicity, Potential ABO incompatibility, Severe infection 0-24h, and genotyping of G6PD, UGT1A1*6 and UGT1A1 promoter with p<0.15 from univariate model. Young maternal age, Oxytocin infusion, sex and G6PD deficiency were all significant in univariable model but were not included in the multivariable model because they were highly correlated with Primigravida, Rupture of membranes ≥ 18h, and G6PD genotyping, respectively. There was no interaction effect between G6PD status (Hemi-Homo/Heterozygote vs WT) and UGT1A1*6 status (homozygote vs non- homozygote), p = 0.171.

[b] Combined PAR% (95%CI) for G6PD non-WT vs WT is 27.2 (19.0–36.0)

Harrell's C statistic for model discrimination = 0.818.

reaching the severe threshold, 3 out 5 males were G6PD*Mahidol hemizygotes (and tested deficient by FST at birth) and 3 out 5 females were G6PD*Mahidol heterozygote (and tested normal by FST); 6 neonates were heterozygote for the UGT1A1*6 allele. Overall, 9/10 neonates with EGA≥38 weeks in the group of severe NH had at least one mutation in either the G6PD or UGT1A1 genes, but only 3 had been diagnosed earlier as having a risk factor. One female term neonate who was heterozygous for G6PD*Mahidol allele had clinical signs of sepsis and severe NH at the day 7 visit (TSB = 1,072 μmol/L) and was referred for exchange transfusion at the local Mae Sot Hospital. Despite receiving 3 exchange transfusions, she died the same day. She was diagnosed with possible ABO incompatibility.

## Analysis of risk factors of "early" and "late" NH in neonates with EGA≥38 weeks

A risk analysis for early and late NH was performed only on 1,124 neonates born with EGA≥38 weeks. Of those, 5.4% (61/1,124) developed NH early (≤48 hours), and 11.4%

developed NH (128/1,124) late (49–168 hours), the remaining 83.2% (935/1,124) did not develop NH. Their characteristics by group are presented in S3 Table.

## Risk factors for early NH

Primigravida and the mother being overweight were independently associated with a 2-fold increased risk of early NH while delayed cord clamping had a protective effect (Table 3 and S4 Table). All mutated G6PD genotypes were associated significantly with an increased risk of developing NH in the first 48 hours of life; G6PD heterozygotes had more than twice the risk of developing early NH as compared to wild type (AHR = 2.61, 95%CI: 1.18–5.77; P = 0.018), while G6PD hemi and homozygous neonates had >9-fold risk as compared to wild type (AHR = 9.18, 95%CI: 4.79–17.59; P<0.001). Combined PAR (95%CI) for mutated G6PD genotypes against wild type genotype was 38.1 (22.1–54.4) %. Mutations in the UGT1A1 gene had no impact on the development of early NH.

## Risk factors for late NH

Maternal haemoglobinopathies, prolonged rupture of membranes, the presence of haematoma at birth, and neonatal sepsis in the first 24 hour of life were all independently associated with an increased risk of late NH (Table 4 and S5 Table). Mutations in the G6PD gene were

**Table 3. Early Neonatal hyperbilirubinaemia (NH) in neonates with EGA≥38 weeks: Uni- and multivariable analysis of genetic factors and other significant risk factors using a mixed effects Cox proportional hazard model clustering by site among neonates who developed NH early (within 48 hours) and neonates who did not develop NH in the first week of life.**

| Characteristics | Univariable analysis | | Multivariable analysis[a] | | PAR |
|---|---|---|---|---|---|
| | HR (95% CI) | p-value | HR (95% CI) | p-value | % (95% CI) |
| **Newborn genotyping** | | | | | |
| G6PD (any mutation) | | | | | |
| WT | Reference | | Reference | | |
| Heterozygote | 2.27 (1.05, 4.91) | 0.038 | 2.61 (1.18, 5.77) | 0.018 | 13.1 (1.7–30.8)[b] |
| Hemi + Homozygote | 8.75 (4.93, 15.54) | <0.001 | 9.18 (4.79, 17.59) | <0.001 | 34.8 (19.8–52.0)[b] |
| UGT1A1*6 | | | | | |
| WT | Reference | | Reference | | |
| Heterozygote | 0.92 (0.51, 1.66) | 0.781 | 1.17 (0.64, 2.17) | 0.608 | |
| Homozygote | 2.49 (0.77, 8.03) | 0.127 | 1.47 (0.43, 5.10) | 0.540 | |
| UGT1A1*28 | | | | | |
| WT (TA6/6) | Reference | | | | |
| Hetero and homozygote (TA6/7+ TA7/7) | 1.17 (0.63, 2.16) | 0.620 | | | |
| **Maternal Characteristics** | | | | | |
| Primigravida (Primipara) | 1.74 (1.05, 2.88) | 0.032 | 2.06 (1.10, 3.88) | 0.024 | 26.4 (3.3–49.3) |
| Overweight | 2.35 (1.41, 3.95) | 0.001 | 2.15 (1.20, 3.88) | 0.011 | 21.6 (4.6–40.9) |
| **Obstetric characteristics** | | | | | |
| Delayed cord clamping | 0.36 (0.20, 0.64) | <0.001 | 0.38 (0.17, 0.84) | 0.018 | |

WT: wild type; HR: Hazard ratio; CI: confidence interval; PAR: population attributable Risk (sum of PAR can be >100% [24])

[a] Adjusted for Primigravida, Overweight, Pre-eclampsia or eclampsia, Oxytocin infusion, Delayed cord clamping, Resuscitation, Presence of haematoma, Sibling with history of jaundice, Potential ABO incompatibility, Positive Coombs test and genotyping of G6PD and UGT1A1*6 with p<0.15 from univariate model. Sex and G6PD deficiency by FST were significant in the univariable model but not included in the multivariable model because they were highly correlated with G6PD genotyping. There was no interaction effect between G6PD status (Hemi-Homo/Heterozygote vs WT) and UGT1A1*6 status (homozygote vs non- homozygote), p = 0.658.

[b] Combined PAR% (95%CI) for G6PD non-WT vs WT is 38.1 (22.1–54.4)

Harrell's C statistic for model discrimination = 0.824.

**Table 4. Late Neonatal hyperbilirubinaemia (NH) in neonates with EGA≥38 weeks: Uni- and multivariable analysis of genetic factors and other significant risk factors using a mixed effects Cox proportional hazard model clustering by site among neonates who developed NH late (49–168 hours) and neonates who did not develop NH in the first week of life.**

| Characteristics | Univariable analysis | | Multivariable analysis[a]: Model A | | PAR |
|---|---|---|---|---|---|
| | HR (95% CI) | p-value | HR (95% CI) | p-value | % (95% CI) |
| **Newborn genotyping** | | | | | |
| G6PD (any mutation) | | | | | |
| WT | Reference | | Reference | | |
| Heterozygote | 2.07 (1.26, 3.41) | 0.004 | 2.16 (1.23, 3.78) | 0.007 | 10.2 (2.2–21.5)[b] |
| Hemi + Homozygote | 3.59 (2.20, 5.85) | <0.001 | 4.40 (2.51, 7.71) | <0.001 | 17.7 (8.7–29.8)[b] |
| UGT1A1*6 | | | | | |
| WT | Reference | | Reference | | |
| Heterozygote | 1.51 (1.03, 2.23) | 0.036 | 1.34 (0.87, 2.06) | 0.179 | |
| Homozygote | 4.46 (2.45, 8.11) | <0.001 | 3.77 (2.01, 7.08) | <0.001 | 7.7 (3.0–15.5)[b] |
| UGT1A1*28 | | | | | |
| WT (TA6/6) | Reference | | Reference | | |
| Hetero and homozygote (TA6/7+ TA7/7) | 0.34 (0.18, 0.65) | 0.001 | 0.40 (0.20, 0.81) | 0.011 | |
| **Maternal Characteristics** | | | | | |
| Haemoglobinopathies | 1.81 (1.10, 2.99) | 0.020 | 1.88 (1.09, 3.26) | 0.024 | 6.6 (0.7–15.3) |
| **Obstetric characteristics** | | | | | |
| Rupture of membranes ≥ 18h | 1.63 (0.90, 2.97) | 0.107 | 2.02 (1.07, 3.82) | 0.030 | 5.9 (0.4–14.7) |
| **Neonatal Characteristics** | | | | | |
| Presence of haematoma | 2.81 (1.52, 5.22) | 0.001 | 2.04 (1.01, 4.11) | 0.047 | 3.8 (0.0–10.5) |
| **Clinical events** | | | | | |
| Severe infection 0-24h | 2.05 (1.08, 3.92) | 0.029 | 2.29 (1.12, 4.67) | 0.023 | 5.4 (0.5–14.0) |

WT: wild type; HR: Hazard ratio; CI: confidence interval; PAR: population attributable Risk (sum of PAR can be >100% [24])

[a]Adjusted for Literacy, Primigravida, Rupture of membranes ≥ 18h, Presence of hematoma, Sgaw Karen ethnicity, Severe infection 0-24h and genotyping of G6PD, UGT1A1*6, UGT1A1 promoter and mother's haemoglobinopathies with p<0.15 from univariate model. Sex and G6PD deficiency by FST were significant in univariable model but not be included in the multivariable model because they were highly correlated with G6PD genotyping. There was no interaction effect between G6PD status (Hemi-Homo/Heterozygote vs WT) and UGT1A1*6 status (homozygote vs non- homozygote), p = 0.995 and between the G6PD status and UGT1A1*28 status, p = 0.718.

[b] Combined PAR% (95%CI) for G6PD non-WT vs WT is 23.6 (12.6–36.1); Combined PAR% (95%CI) for G6PD non-WT (vs WT) or UGT1A1*6 homozygote (vs heterozygote +WT) or mother with haaemoglobinopathies is 34.9 (21.9–47.8)

Harrell's C statistic for model discrimination = 0.753.

associated with significant risk of late NH (AHR = 2.16, 95% CI: 1.23–3.78; P = 0.007 for heterozygotes and AHR = 4.40, 95% CI: 2.54–7.71; P<0.001, for hemi and homozygotes) with combined PAR% (95%CI) of 23.6 (12.6–36.1). Homozygotes for UGT1A1*6 had an almost 4-fold increase in risk of late NH (AHR = 3.77, 95%CI: 2.01–7.08; P<0.01- PAR = 7.7%) while UGT1A1*28 in the promoter had a protective effect compared to G6PD wild type genotype (AHR = 0.40, 95%CI: 0.20–0.81; P = 0.011). When risk factors were compared in neonates developing NH between 72 hours and one week of life and those who did not develop NH in the first week of life (S6 Table), the UGT1A1*6 homozygotes had an even higher AHR of 7.78 (95%CI: 3.68–16.47; P<0.001).

## Analysis of TSB levels

A comparison of TSB levels at 24h (±4h), 48h(±4h), 72h(±4h) and 168h(±4h) follow-up in neonates with EGA≥38 weeks who did not need phototherapy, or before they had received it,

is shown in Fig 3 and S7 Table. While the physiologic increase in TSB levels between 24h and 48h was slightly more pronounced in neonates with G6PD mutations, TSB levels after 48 hours increased substantially in G6PD wild type neonates with homozygote UGT1A1*allele. Neonates with EGA≥38 weeks who are G6PD normal are usually considered low risk and tend to be discharged early from the clinic. There was no interaction between G6PD mutated and UGT1A1*6 homozygous genotypes on TSB levels over time (slope of regression (95%CI): -8.8 (-38.7, 21.0); P = 0.56) although the number of neonates with both conditions was very small (4, 3, 2 and 1 G6PD mutated and UGT1A1*6 homozygotes per time point, S7 Table).

## Analysis of risks factors for prolonged jaundice in the first month of life

A total of 1,596 neonates (185 with EGA<38 weeks and 1,411 with EGA≥38 weeks) with at least one follow-up visit after the first week of life were included in the analyses.

Overall, over a third (615/1,596, 38.5%) of the neonates followed up between 14 days and one month of life had visible jaundice at one follow-up visit; neonates with lower EGA (<38 weeks) were more likely to develop prolonged jaundice compared to neonates with EGA≥38 weeks (107/185 vs 507/1,411, p<0.001).

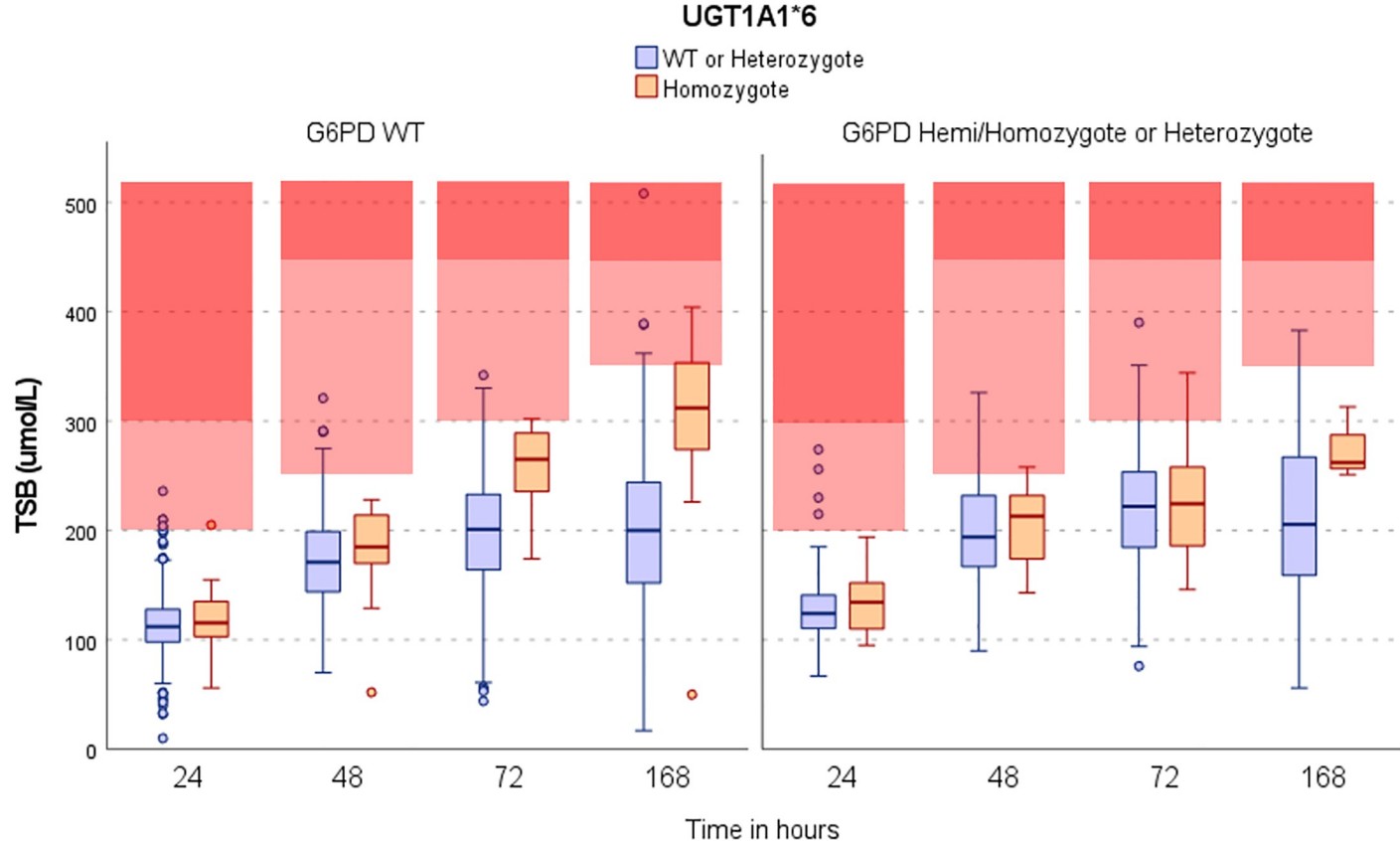

**Fig 3. Total serum bilirubin concentrations in neonates with EGA≥38 weeks: UGT1A1*6 genotype (homozygote vs non-homozygote) in G6PD hemi-homo/ heterozygote group and wild type group.** TSB = Total Serum Bilirubin; Boxes represent inter quartile ranges; middle horizontal lines are medians Red areas represent TSB values at which phototherapy (light red) and exchange transfusion (darker red) should be provided based on neonates' age (NICE guideline). One data point (TSB = 1072µmol/L) at 168h in the right panel among UGT1A1*6 WT-heterozygote is not shown.

A large proportion of neonates with prolonged jaundice (64.7%, 397/614) did not have NH in the first week of life. Of the neonates who had NH in the first week of life, 60.6% (217/358) had also prolonged jaundice.

Most neonates were exclusively breastfed, both among those who had prolonged jaundice (94/107, 87.9% in EGA<38 weeks and 497/507, 98.0% in EGA≥38 weeks) and those who did not have it (73/78, 93.6% in EGA<38 weeks and 890/904, 98.5% in EGA≥38 weeks).

Among jaundiced neonates, further clinical and laboratory investigations were done for 80 (74.8%) neonates with EGA <38 weeks and 359 (70.8%) neonates with EGA≥38 weeks. None of the neonates with prolonged jaundice were diagnosed with intra or extrahepatic disease.

EGA <38 weeks and presence of haematoma at birth were the only clinical factors which were significantly associated with an increased risk of developing prolonged jaundice (Table 5). Neonates with G6PD hemi and homozygous genotypes had more than 2-fold increased risk (95%CI:1.3–3.2, p = 0.002) of having prolonged jaundice in their first month of life and those carriers of UGT1A1*6 both in heterozygosity and homozygosity had a risk of about 1.6 (95%CI:1.2–2.0, p = 0.001) and 3.6 (95%CI:1.8–6.9, p<0.001) respectively. The proportion of newborns with prolonged jaundice at each follow-up visit according to G6PD and UGT1A1 genotypes is shown in S9 Table.

Of the 1,596 neonates, the majority (76.6%, 1,222) had a full follow-up with three completed visits at week 2, week 3 and one month of age. Among those with a full follow-up, 761 (62.3%) never had visible yellow skin. The majority of neonates with prolonged jaundice were visibly jaundiced during the first 3 weeks (231/462) and one month (166/462) of life; only a small minority were jaundiced only at week 2 (65/462). Analysis of risk factors for duration of prolonged jaundice (Table 6) identified gestational age and positive Coomb's test as significant factors in addition to the heterozygous and homozygous UGT1A1*6 genotypes (Incidence rate ratio[95%CI] of 1.4 [1.1–1.7], p = 0.001 and 2.4[1.5–3.9], p<0.001 respectively) and G6PD deficiency (hemi and homozygous genotypes, IRR[95%CI] = 1.6[1.2–2.3], p = 0.003).

Total and indirect bilirubin levels were analysed in neonates who had visible yellow skin discoloration at the 3-weeks follow-up visit. Bilirubin analysis results and genotype were available in 87.5% (386/441) of jaundiced neonates at that visit. Mean [SD] total and indirect bilirubin levels were 156.2[58.2] μmol/L and 138.8 [56.1] μmol/L in neonates with EGA <38 weeks with similar values in neonates with EGA>38 weeks; 144.6 [61.3] μmol/L and 128.4 [60.0] μmol/L respectively. Indirect bilirubin levels (i.e. unconjugated bilirubin) were similar among neonates with different G6PD genotypes but were significantly higher in 132 neonates with UGT1A1*6 heterozygous genotype (143.6 [60.9] μmol/L) and 23 neonates with homozygous genotype (182.5 [62.4] μmol/L) as compared with the 230 neonates with UGT1A1 wild type genotype (117.3 [53.8] μmol/L; P<0.001). No differences were observed among neonates with different genotypes in the UGT1A1 promoter.

## Discussion

In this population living along the Thailand-Myanmar border low EGA was the main risk factor for NH in the first week of life (PAR[95%CI] = 61.7 [54.2–68.6]%); Among neonates with EGA≥38 weeks, the analysed genetic risk factors had a combined PAR (95%CI) of 38.1 (22.1–54.4) % for early NH and 34.9 (21.9–47.8) % for late NH. Currently available tests in most low-resource settings do not identify all neonates at risk, especially term neonates who are often discharged around two days of life. Identification of risk factors at birth for a "late" increase in bilirubinaemia levels is particularly important because these neonates may not be able to access required medical care or may access it too late.

**Table 5. Prolonged jaundice (between day 14–35 days): Uni- and multivariable analysis of potential risk factors and genotyping using mixed effects logistic model clustering by site.**

| Characteristics | Neonates with prolonged Jaundice, n (%) (N = 615) | Neonates without prolonged Jaundice, n (%) (N = 981) | Univariable analysis | | Multivariable analysis [a] | | PAR |
|---|---|---|---|---|---|---|---|
| | | | OR (95% CI) | p-value | OR (95% CI) | p-value | % (95% CI) |
| **Newborn genotyping** | | | | | | | |
| G6PD (any mutation) [b] | (N = 573) | (N = 914) | | | | | |
| WT | 465 (81) | 772 (84) | Reference | | Reference | | |
| Heterozygote | 56 (10) | 92 (10) | 1.00 (0.69, 1.44) | 0.998 | 1.00 (0.69, 1.46) | 0.982 | |
| Hemi + Homozygote | 52 (9) | 50 (6) | 1.93 (1.27, 2.95) | 0.002 | 2.06 (1.31, 3.24) | 0.002 | 6.8 (2.1–13.4) |
| UGT1A1*6 [b] | (N = 555) | (N = 914) | | | | | |
| WT | 356 (62) | 665 (73) | Reference | | Reference | | |
| Heterozygote | 187 (33) | 233 (25) | 1.61 (1.26, 2.05) | <0.001 | 1.57 (1.22, 2.02) | 0.001 | 14.0 (5.9–22.6) |
| Homozygote | 29 (5) | 16 (2) | 3.50 (1.84, 6.67) | <0.001 | 3.57 (1.84, 6.91) | <0.001 | 7.4 (2.5–15.5) |
| UGT1A1*28 | (N = 489) | (N = 713) | | | | | |
| WT (TA6/6) | 366 (75) | 567 (80) | Reference | | | | |
| Hetero and homozygote (TA6/7 + TA7/7) | 123 (25) | 146 (20) | 1.23 (0.92, 1.63) | 0.156 | | | |
| **Maternal Characteristics** | | | | | | | |
| Young maternal age (≤20 y) | 191 (31) | 245 (25) | 1.30 (1.03, 1.63) | 0.028 | 1.24 (0.96, 1.59) | 0.097 | |
| Illiterate (cannot read) | 200 (33) | 363 (37) | 0.87 (0.70, 1.08) | 0.214 | | | |
| Smoking | 53 (9) | 90 (9) | 0.93 (0.65, 1.35) | 0.718 | | | |
| Primigravida (Primipara) | 233 (38) | 321 (33) | 1.23 (0.99, 1.53) | 0.059 | | | |
| Overweight | 146/596 (25) | 237/967 (25) | 1.09 (0.85, 1.39) | 0.505 | | | |
| Pre-eclampsia or eclampsia | 15 (2) | 24 (2) | 1.08 (0.55, 2.11) | 0.833 | | | |
| Haemoglobinopathies | 42 (7) | 80 (8) | 0.88 (0.59, 1.31) | 0.535 | | | |
| **Obstetric characteristics** | | | | | | | |
| Rupture of membranes ≥ 18h | 45/564 (8) | 61/900 (7) | 1.08 (0.72, 1.64) | 0.698 | | | |
| Oxytocin infusion | 70/606 (12) | 108/958 (11) | 1.07 (0.77, 1.49) | 0.679 | | | |
| Delayed cord clamping | 494 (80) | 775 (79) | 1.03 (0.79, 1.33) | 0.836 | | | |
| **Neonatal Characteristics** | | | | | | | |
| Gestational age <38 weeks | 108 (18) | 77 (8) | 2.77 (2.00, 3.84) | <0.001 | 2.79 (1.95, 3.98) | <0.001 | 17.2 (9.9–25.7) |
| ≥38 weeks | 507 (82) | 904 (92) | Reference | | Reference | | |
| Resuscitation | 17/598 (3) | 30/942 (3) | 0.87 (0.47, 1.61) | 0.649 | | | |
| Presence of haematoma | 31/607 (5) | 28/ 968 (3) | 1.85 (1.08, 3.19) | 0.026 | 1.95 (1.10, 3.44) | 0.022 | 3.4 (0.4–8.3) |
| Sgaw Karen ethnicity | 249/599 (42) | 363/955 (38) | 1.33 (1.04, 1.70) | 0.021 | 1.20 (0.92, 1.56) | 0.187 | |
| Male sex | 354 (58) | 467 (48) | 1.50 (1.21, 1.85) | <0.001 | * | | |
| Small for gestational age | 119/610 (20) | 190/975 (20) | 0.96 (0.74, 1.25) | 0.754 | | | |
| Sibling with history of jaundice | 68 (11) | 107 (11) | 1.01 (0.73, 1.41) | 0.939 | | | |
| Use of naphthalene for storing the clothes | 33 (5) | 51 (5) | 0.88 (0.55, 1.40) | 0.590 | | | |
| G6PD deficiency (by FST) | 538 (9) | 57 (6) | 1.98 (1.33, 2.95) | 0.001 | * | | |
| Potential ABO incompatibility | 94 (15) | 146 (15) | 1.03 (0.77, 1.38) | 0.843 | | | |

(*Continued*)

**Table 5.** (Continued)

| Characteristics | Neonates with prolonged Jaundice, n (%) (N = 615) | Neonates without prolonged Jaundice, n (%) (N = 981) | Univariable analysis | | Multivariable analysis [a] | | PAR |
|---|---|---|---|---|---|---|---|
| | | | OR (95% CI) | p-value | OR (95% CI) | p-value | % (95% CI) |
| Positive Coombs test | 14/543 (3) | 32/869 (4) | 0.62 (0.32, 1.20) | 0.156 | | | |
| **Clinical events** | | | | | | | |
| Severe infection 0-24h | 30 (5) | 38 (4) | 1.34 (0.81, 2.23) | 0.258 | | | |
| Weight loss ≥7% at 24h [12-30h] of life | 13/610 (2) | 20/974 (2) | 0.87 (0.42, 1.80) | 0.714 | | | |
| HCT at 24h [12-30h] of life | mean (SD) 58.1 (6.8) | mean (SD) 57.8 (7.2) | 1.69 (0.31, 9.12) (Per 10-unit increment) | 0.540 | | | |
| Polycythaemia (HCT > = 70%) at 24 [12-30h] of life | 45 (7) | 73 (7) | 1.17 (0.78, 1.76) | 0.443 | | | |

WT: wild type; OR: Odds ratio; CI: Confidence interval; PAR: population attributable Risk (sum of PAR can be >100% [24])

[a] Adjusted for Young maternal age, Gestational age group, Presence of hematoma, Sgaw Karen ethnicity, genotyping of G6PD and UGT1A1 with p<0.15 from univariate model.

[b] There was no significant interaction effects between G6PD status (Hemi-Homo/Heterozygote vs WT) and UGT1A1 status (homozygote vs non- homozygote) on prolonged jaundice in the first month of life (p = 0.468)

* Gender and G6PD deficiency result from FST were not included in the model because they were highly correlated with G6PD genotyping.

G6PD deficiency was strongly associated with both early and late NH. Hemizygotes and homozygotes had adjusted risks (AHR) of more than 9 for early and more than 4 for late NH. G6PD heterozygotes also had an increased risk of more than twice that in the remaining population [25–27]. G6PD Mahidol is the main genotype accounting for nearly 90% of G6PD deficiency. The allele frequency varies among the different ethnic groups but averages approximately 10%. Currently available qualitative point-of-care tests have moderate sensitivity for identifying deficient newborns (8.7% of deficient newborns missed) and cannot identify females with intermediate phenotypes [28]. In this series, 98% (121/123) of G6PD heterozygous neonates were classified as phenotypically normal using the rapid FST screening test and could not be diagnosed as being at increased risk during post-natal care.

In this population, mutations in the bilirubin conjugating enzyme UGT1A1 were described for the first time. UGT1A1*6 allele was common (prevalence ranging from 12% to 21% in the major ethnic groups) and was associated specifically with late NH, which developed more commonly than early NH in term neonates. UGT1A1*6 homozygotes had a 3-fold increased risk of NH in the first week of life, in particular after 2–3 days of life. Increased risk of NH in UGT1A1*6 was first reported in Japan over 20 years ago [29] and has been observed elsewhere in East Asian countries (Taiwan [30]; Malaysia [31]; Thailand [32]), although in certain contexts the increased risk in neonates with the mutation was only observed in association with large neonatal body weight loss [33]. NH which develops after discharge from birth centres in newborns with higher EGA and no obvious risk factors represents a clinical challenge in low resource settings and, in particular, in migrant populations where access and medical follow-up cannot be provided easily [34]. Levels of TSB observed in neonates homozygous for UGT1A1*6 at 48h (i.e. roughly around the time of discharge) were only slightly elevated as compared to UGT1A1 wild type and would not have justified extended observation. Since reduced activity of UGT1A1 cannot be identified by a simple laboratory test, a genotyping test for the UGT1A1*6 mutation in either the expectant mother or the neonates at birth, may be warranted especially in the ethnic groups with the high allele frequencies. Parental education

**Table 6. Uni- and multivariable analysis of genetic factors and other significant risk factors for duration of prolonged jaundice using mixed effects negative binomial model clustering by site.**

| Characteristics | Univariable analysis | | Multivariable analysis [a] | |
|---|---|---|---|---|
| | IRR (95% CI) | p-value | IRR (95% CI) | p-value |
| **Newborn genotyping** | | | | |
| G6PD (any mutation) [b] | | | | |
| WT | Reference | | Reference | |
| Heterozygote | 0.99 (0.73, 1.33) | 0.929 | 0.95 (0.70, 1.30) | 0.766 |
| Hemi + Homozygote | 1.49 (1.09, 2.04) | 0.012 | 1.63 (1.18, 2.26) | 0.003 |
| UGT1A1*6 [b] | | | | |
| WT | Reference | | Reference | |
| Heterozygote | 1.39 (1.15, 1.69) | 0.001 | 1.40 (1.14, 1.71) | 0.001 |
| Homozygote | 2.22 (1.42, 3.47) | <0.001 | 2.42 (1.51, 3.86) | <0.001 |
| UGT1A1*28 | | | | |
| WT (TA6/6) | Reference | | | |
| Hetero and homozygote (TA6/7+ TA7/7) | 1.13 (0.90, 1.42) | 0.295 | | |
| **Neonatal Characteristics** | | | | |
| Gestational age | | | | |
| <38 weeks | 1.86 (1.47, 2.35) | <0.001 | 1.96 (1.52, 2.54) | <0.001 |
| ≥38 weeks | Reference | | Reference | |
| Sgaw Karen ethnicity | 1.32 (1.09, 1.61) | 0.005 | 1.22 (0.99, 1.51) | 0.065 |
| Positive Coombs test | 0.63 (0.37, 1.06) | 0.083 | 0.58 (0.34, 0.99) | 0.044 |

WT: wild type; IRR: Incidence rate ratio; CI: Confidence interval

[a] Adjusted for Gestational age group, Sgaw Karen ethnicity, Positive Coombs test, genotyping of G6PD and UGT1A1 with p<0.15 from univariate model. Gender and G6PD deficiency result from FST were not included in the model because they were highly correlated with G6PD genotyping.

[b] There was no significant interaction between G6PD status (Hemi-Homo/Heterozygote vs WT) and UGT1A1 status (homozygote vs non- homozygote) on duration of prolonged jaundice in the first month of life (p = 0.735)

about signs of NH after discharge from hospital remains of paramount importance and is feasible in low-resource settings; while neonates with undiagnosed UGT1A1*6 mutations might be numerically few, their individual risk is high.

Increased number of TA repeats in the promoter (UGT1A1*28; a common cause of Gilbert's syndrome) was found in 12% of the newborns and was associated with lower risk of NH. Two meta-analyses conducted in 2015 and 2020 [35, 36] showed a large variability in risk of NH for each variant across the 34 included studies. Overall, the UGT1A1*6 allele was associated with a larger risk as compared to allele *28 (the latter found mostly in African populations). Studies in East Asia where both alleles were analysed in the same newborn population ((Malaysia [31]; Vietnam [37]; Taiwan [30] and China [38]) provided very similar results to those observed here. In these studies, UGT1A1*6 allele had a higher population frequency as compared to allele UGT1A1*28 and was associated with increased risk of NH as opposed to allele UGT1A1*28. This suggests different contributions to increased risk by different mutations on the UGT1A1 gene according to their prevalence.

A novel result was the impact of beta-thalassemia trait and HbE of the mother on the increased risk (AHR = 1.88, 95%CI 1.09–3.26) of developing late NH (after 48 hours of life). Hypothesising increased neonatal anaemia in these cases, we analysed haematocrits at 24 hours of life (S9 Table) but no differences in infants' haematocrit values were observed.

Prolonged jaundice was common and mostly uncomplicated in this population of mainly breastfed neonates, an association described extensively in the literature since the 1960s [39].

There are different recommendations regarding the required investigations in case of prolonged neonatal jaundice [40, 41] in order to exclude potential treatable causes (including sepsis, urine tract infection, hypothyroidism, metabolic and liver disease- mainly congenital biliary atresia). In this study, in addition to lower EGA, mutations in the G6PD and UGT1A1 genes were the major risk factors for prolonged jaundice; total and especially indirect bilirubin levels were significantly elevated in neonates with the UGT1A1*6 allele. Commonly seen "breast milk jaundice" might indeed have a genetic component due to this trait [42].

The current analysis has some limitations. Other genetic traits not analysed here presumably contributed to increased risk. Other UGT1A1 variants associated to increased risk in Asian population [43] have not been investigated here. Variation in expression of the HMOX gene which encodes for heme oxygenase, the enzyme responsible for transformation of heme into biliverdin, has been shown to be associated with increased risk of NH [38]. Analysis of the gene promoter in the local adult Karen and Burman population, showed a high degree of polymorphism [44]. Mutations on the alpha-globin genes were not analysed in this cohort but are common in the population (around 25% carrier; [17]) and might play a role on the onset of NH [45, 46].

## Future perspectives

In conclusion, the high risk of NH in the first week of life in this cohort of Karen and Burman neonates was mainly a result of lower EGA. This has many aetiologies but, in low resource settings, infections are an important and preventable cause [47]. For neonates with EGA≥38 weeks, some actionable risk factors were identified such as excess maternal weight gain [48], prolonged rupture of membranes, trauma at birth and neonatal sepsis. The analysis showed that delayed cord clamping, which is an inexpensive practice associated with multiple benefits for the newborn, also reduces risk of NH in this population with multiple risk factors [49]. Genetic risk factors, common in this population, play a large role in neonatal jaundice, including the severe forms, as seen in other low-resource settings [50]. Improved diagnostics are urgently needed and different screening strategies should be considered in populations with a high prevalence of these traits. Genotyping of expectant mothers might prove cost-effective in settings where high-throughput techniques are widely available. For example, this would be useful for ruling-out heterozygosity for UGT1A1*6 allele in the mother (necessary for homozygosity in newborn) or planning for extended monitoring of bilirubin levels after birth. In rural and low-resources settings, in lack of simple and cheap genetic tests for UGT1A1 (such as a LAMP-based PCR test), continued neonatal bilirubin monitoring (where possible) and education on signs of NH remain the only feasible approaches. For G6PD deficiency, easy to use quantitative point-of-care tests (in place of qualitative tests) able to identify both deficient and intermediate phenotypes would represent a cost-effective tool to provide appropriate care in male and female neonates at risk. One such test has been recently evaluated with good results in this setting among newborns (Bancone, in preparation) and few more are in the late stage of development.

## Supporting information

**S1 Table. Primers, PCR conditions, restriction enzyme and band interpretation for genotyping of allele *6 and promoter of the human UGT1A1 gene.**
(DOCX)

**S2 Table. Uni- and multivariable analysis of potential risk factors and genotyping for developing NH in the first week of life using mixed effect Cox proportional hazard model**

**clustering by site.**
(DOCX)

**S3 Table. Characteristics of neonates with EGA $\geq$ 38 weeks who developed NH within 48 hours, between 49 and 168 hours and not develop NH in the first week of life.**
(DOCX)

**S4 Table. Early NH: Uni- and multivariable analysis of potential risk factors using a mixed effects Cox proportional hazard model clustering by site among neonates $\geq$ 38 weeks who developed NH early (within 48 hours) and neonates who did not develop NH in the first week of life.**
(DOCX)

**S5 Table. Late NH: Uni- and multivariable analysis of potential risk using mixed effects Cox proportional hazard model clustering by site among neonates $\geq$ 38 weeks who developed NH late (49–168 hours) and neonates who did not develop NH in the first week of life.**
(DOCX)

**S6 Table. Uni- and multivariable analysis of potential risk factors and genotyping for developing NH in the first week of life using mixed effects Cox proportional hazard model clustering by site among neonates who develop NH after 72 hours of life and neonates who did not develop NH in the first week of life.**
(DOCX)

**S7 Table. Median total serum bilirubin concentrations over time by genotype.**
(DOCX)

**S8 Table. Proportion (%) of newborns with prolonged jaundice at each follow-up week according to genotype.**
(DOCX)

**S9 Table. Haematocrits tested at 24 hours in neonates born from mothers with different haemoglobin types.**
(DOCX)

## Acknowledgments

The authors wish to thank all the mothers (and neonates) for the collaboration and patience. They also want to thank the clinical and laboratory SMRU staff involved in the study for their hard work and dedication. Thanks also to the paediatric staff at Mae Sot General Hospital for their care of neonates requiring exchange transfusion.

## Author Contributions

**Conceptualization:** Germana Bancone, Francois Nosten, Rose McGready, Verena I. Carrara.

**Data curation:** Germana Bancone, Gornpan Gornsawun, Pimnara Peerawaranun, Penporn Penpitchaporn, Moo Kho Paw, Day Day Poe, December Win, Naw Cicelia, Laypaw Archa-suksan, Laurence Thielemans, Verena I. Carrara.

**Formal analysis:** Pimnara Peerawaranun, Mavuto Mukaka.

**Funding acquisition:** Laurence Thielemans.

**Investigation:** Germana Bancone, Gornpan Gornsawun, Pimnara Peerawaranun, Penporn Penpitchaporn, Moo Kho Paw, Day Day Poe, December Win, Naw Cicelia, Mavuto Mukaka, Laypaw Archasuksan, Laurence Thielemans, Nicholas J. White, Rose McGready, Verena I. Carrara.

**Methodology:** Germana Bancone, Gornpan Gornsawun, Penporn Penpitchaporn, Day Day Poe, December Win, Naw Cicelia, Laypaw Archasuksan, Laurence Thielemans, Rose McGready, Verena I. Carrara.

**Software:** Mavuto Mukaka.

**Supervision:** Germana Bancone, Moo Kho Paw, Mavuto Mukaka, Francois Nosten, Nicholas J. White, Rose McGready, Verena I. Carrara.

**Validation:** Laurence Thielemans.

**Visualization:** Nicholas J. White.

**Writing – original draft:** Germana Bancone.

**Writing – review & editing:** Germana Bancone, Francois Nosten, Nicholas J. White, Rose McGready, Verena I. Carrara.

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
