## [Decision Letter · Decision Letter 0]

6 Apr 2022

PGPH-D-22-00231

Contribution of genetic factors to high rates of neonatal hyperbilirubinaemia on the Thailand-Myanmar border

Dear Germana Bancone,

Thank you for submitting your manuscript to PLOS Global Public Health. After careful consideration, we feel that it has merit but does not fully meet PLOS Global Public Health’s publication criteria as it currently stands. Therefore, we invite you to submit a revised version of the manuscript that addresses the points raised during the review process.

We look forward to receiving your revised manuscript.

Kind regards,

Collins Otieno Asweto, PhD

Academic Editor

Journal Requirements:

1. Your co-authors, Gornpan Gornsawun (gornpan@shoklo-unit.com), Penporn Penpitchaporn (penporn@shoklo-unit.com), Moo Kho Paw (mookhopaw@shoklo-unit.com), Day Day Poe (daydaypoe@shoklo-unit.com), December Win (december@shoklo-unit.com), Naw Cicelia (nawcicelia@shoklo-unit.com), Mavuto Mukaka (mavuto@tropmedres.ac), Laypaw Archasuksan (laypaw@shoklo-unit.com), Laurence Thielemans (thielemans.laurence@gmail.com), Francois Nosten (francois@tropmedres.ac), and Rose Mc Gready (rose@shoklo-unit.com), have not confirmed authorship of the manuscript. We have resent them the authorship confirmation email; however please check that the above email address for them is correct and follow up personally to ensure they confirm. Please note that we cannot pass your manuscript to Production until we have received confirmations from all co-authors. 

2. In the online submission form, you indicated that “Data are available from MORU Tropical Health Network upon request from this link: https://www.tropmedres.ac/units/moru-bangkok/bioethics-engagement/data-sharing”. All PLOS journals now require all data underlying the findings described in their manuscript to be freely available to other researchers, either 1. In a public repository, 2. Within the manuscript itself, or 3. Uploaded as supplementary information.

3. We ask that a manuscript source file is provided at Revision. Please upload your manuscript file as a .doc, .docx, .rtf or .tex. If you are providing a .tex file, please upload it under the item type ‘LaTeX Source File’ and leave your .pdf version as the item type ‘Manuscript’.

4. Please provide separate figure files in .tif or .eps format only and ensure that all files are under our size limit of 20MB.

5. We have noticed that you have uploaded supporting information but you have not included a list of legends.  Please add a full list of legends for all supporting information files (including figures, table and data files) after the references list.

Additional Editor Comments (if provided):

Reviewers' comments:

Reviewer's Responses to Questions

**Comments to the Author**

1. Does this manuscript meet PLOS Global Public Health’s publication criteria? Is the manuscript technically sound, and do the data support the conclusions? The manuscript must describe methodologically and ethically rigorous research with conclusions that are appropriately drawn based on the data presented.

Reviewer #1: Yes

Reviewer #2: Yes

2. Has the statistical analysis been performed appropriately and rigorously?

Reviewer #1: Yes

Reviewer #2: Yes

3. Have the authors made all data underlying the findings in their manuscript fully available (please refer to the Data Availability Statement at the start of the manuscript PDF file)?

Reviewer #1: Yes

Reviewer #2: Yes

4. Is the manuscript presented in an intelligible fashion and written in standard English?

Reviewer #1: Yes

Reviewer #2: Yes

5. Review Comments to the Author

Reviewer #1: Reviewer comment

Line 34-35: Increased red blood cell turn-over and immaturity of hepatic glucuronidation both contribute. Revise this statement in terms of language structure

Line 36: define ca

Line 46: state specific haemoglobinopathies

Line 49: define EGA

Lines 38, 43: state specific mutations identified to be associated with NH

Line 70: for reader comprehension, define Viangchan (871G>). This is because the resulting amino acid was not indicated

Lines 54-94: most references are outdated. Provide current references and current information

Line 135, there was an open bracket but not a close bracket. Insert it appropriately

Line 224-225: it is not clear why authors excluded twins from the study. explanation to this decision is expected

Line 285: Tables 2-6, define WT

Reviewer #2: Thank you for the opportunity to review this paper. Congratulations to the team and also apology for delay in reviewing.

Several comments/suggestion:

- Though the title is self explanatory, in the abstract it would be beneficial to add one line stating the aim of the study, and also in method to mention the study design

- In method: please specify the phototherapy used - because in many Asian communities, the tradition to sunbath newborn is quite common. How does the cultural factors are taken into account into the study procedures?

- Please, specify why clustering was important to be taken into account in each model

- Results: very thorough and detailed, use of table is good to visualize what otherwise lost in the dense narrative. The use of PAR as well is very appropriate to express the relative contribution of the studies variables. What were the main assumptions used and why the mixed effect binomial model was used for the prolonged icterus.

- Discussion; The limitations mentioned only other genetic factors that were not investigated in this study, is it because the study was conducted back in 2016-17?

- What are the recommendations that applies to this setting? Would it be feasible and desirable to As said in line 539 on availability of sequencing, do the authors refer to high income countries?

- The conclusion also mentions diagnosis of G6PD, but availability of the desired product (quantitative) seems to be sub optimal, is there any development in the pipeline worth mentioning?

6. PLOS authors have the option to publish the peer review history of their article (what does this mean?). If published, this will include your full peer review and any attached files.

**Do you want your identity to be public for this peer review?** For information about this choice, including consent withdrawal, please see our Privacy Policy.

Reviewer #1: **Yes: **Dr Enoch Aninagyei

Reviewer #2: No

---

## [Editor Report · Decision Letter 1]

5 May 2022

Contribution of genetic factors to high rates of neonatal hyperbilirubinaemia on the Thailand-Myanmar border

PGPH-D-22-00231R1

Dear Dr. Bancone,

We are pleased to inform you that your manuscript 'Contribution of genetic factors to high rates of neonatal hyperbilirubinaemia on the Thailand-Myanmar border' has been provisionally accepted for publication in PLOS Global Public Health.

Best regards,

Collins Otieno Asweto, PhD

Academic Editor
